# Validating the German Short Basic Psychological Need Satisfaction and Frustration Scale in Individuals with Depression

**DOI:** 10.3390/healthcare11030412

**Published:** 2023-01-31

**Authors:** Andreas Heissel, Alba Sanchez, Anou Pietrek, Theresa Bergau, Christiane Stielow, Michael A. Rapp, Jolene Van der Kaap-Deeder

**Affiliations:** 1Social and Preventive Medicine, Department of Sports and Health Sciences, Intra-Faculty Unit “Cognitive Sciences”, Faculty of Human Science, and Faculty of Health Sciences Brandenburg, Research Area Services Research and e-Health, University of Potsdam, 14469 Potsdam, Germany; 2Social and Preventive Medicine, Department of Sports and Health Sciences, Faculty of Human Science, University of Potsdam, 14469 Potsdam, Germany; 3Department of Psychology, Norwegian University of Science and Technology, 7034 Trondheim, Norway

**Keywords:** basic psychological need frustration, need satisfaction, mental health, ill-being, depression

## Abstract

Satisfaction and frustration of the needs for autonomy, competence, and relatedness, as assessed with the 24-item Basic Psychological Need Satisfaction and Frustration Scale (BPNSFS), have been found to be crucial indicators of individuals’ psychological health. To increase the usability of this scale within a clinical and health services research context, we aimed to validate a German short version (12 items) of this scale in individuals with depression including the examination of the relations from need frustration and need satisfaction to ill-being and quality of life (QOL). This cross-sectional study involved 344 adults diagnosed with depression (M_age_ (SD) = 47.5 years (11.1); 71.8% females). Confirmatory factor analyses indicated that the short version of the BPNSFS was not only reliable, but also fitted a six-factor structure (i.e., satisfaction/frustration X type of need). Subsequent structural equation modeling showed that need frustration related positively to indicators of ill-being and negatively to QOL. Surprisingly, need satisfaction did not predict differences in ill-being or QOL. The short form of the BPNSFS represents a practical instrument to measure need satisfaction and frustration in people with depression. Further, the results support recent evidence on the importance of especially need frustration in the prediction of psychopathology.

## 1. Introduction

The recent body of work in the field of Self-Determination Theory (SDT, [1]) suggests that SDT can contribute to our understanding of the onset and course of psychopathological phenomena, such as depression. The Basic Psychological Needs Theory (BPNT), a central sub-theory within SDT, is especially relevant and states that there exist three innate, universal psychological needs that act as essential nutrients for autonomous motivation, optimal functioning, and well-being: autonomy (i.e., having a sense of volition and choice), competence (i.e., experiencing a sense of mastery), and relatedness (i.e., feeling connected to important others) [2].

Research within BPNT that initially focused on the benefits of need satisfaction for individuals’ thriving demonstrated the association of need satisfaction with a variety of well-being indicators (e.g., vitality, self-esteem, and life satisfaction) (see [3] for an overview). An increasing focus of research has more recently investigated the dysfunctional side of human development, focusing on the concept of need frustration, which is referring to the active obstruction and undermining of the basic psychological needs for autonomy (i.e., feelings of pressure), competence (i.e., experiencing oneself as a failure), and relatedness (i.e., feeling socially excluded) [3,4]. The assumption that unmet personal needs lead to health costs and psychological distress is also present in other research areas such as the psychological pain theory (“psychache”) ([5]; also see [6]). Conceptually, as well as empirically, the distinction between need satisfaction and need frustration is justified, as these constructs are not only negatively related to one another but also display different outcomes and antecedents [4,7]. Additionally, both experiences are asymmetrically related to one another, such that the absence of need satisfaction (e.g., perceiving limited choice) does not necessarily mean the presence of need frustration (e.g., feeling forced to engage in a certain activity), but the presence of need frustration always implies the absence of need satisfaction [4].

Previous research indeed yielded empirical support for the important distinction between need satisfaction and need frustration, with need satisfaction being more strongly related to indicators of optimal development (e.g., life satisfaction), whereas need frustration is imperative in the prediction of ill-being and even psychopathology [7]. To illustrate, Bartholomew et al. (2011) [8] initially confirmed the additional costs associated with need frustration within the sport context showing that experiences of need frustration, compared to low need satisfaction, are more strongly associated with ill-being (e.g., burn-out and depression). Results of further studies focusing on different age groups and diverse life domains (e.g., exercise, education, and work) pointed in a similar direction by demonstrating significant associations between need frustration and diverse indicators of ill-being such as depressive symptoms [9], poor sleep (e.g., [10]), and rigid and obsessive behavior [11]. In line with these studies, need frustration has also been found to account for the comorbidity between symptoms of psychopathology, attesting to the transdiagnostic role of the needs [9,12]. That is, besides an increasing number of studies showing the importance of need frustration in diverse symptoms of psychopathology, need frustration also accounts for why certain symptoms coincide and thereby represents a common mechanism in psychopathology.

The effects of basic psychological needs are assumed to be universal, regardless of individuals’ gender, age, and culture. Indeed, previous research has shown the beneficial and detrimental effects of need satisfaction and need frustration, respectively, across cultures, ages, personalities, etc. [13,14,15]. However, it is currently unclear whether these effects also generalize to clinical samples, specifically to individuals with mental disorders such as depression. On a theoretical level, need frustration is expected to also play a crucial role in the prediction of clinical forms of depression [7]. That is, in SDT, it is assumed that permanent thwarting of the basic psychological needs hinders people’s innate propensities and thus leads to a loss of motivation, even reaching a state of amotivation, reflecting discouragement and helplessness (i.e., core characteristics of depression). Such a lack of motivation is one possible mechanism that can then result in decreases in well-being [16]. Although several studies employing community samples have shown need frustration to relate to depressive symptoms (e.g., [9]), research in samples with mental disorders such as depression is generally lacking.

Currently, the most widely-used instrument to assess need satisfaction and need frustration is the Basic Psychological Need Satisfaction and Frustration Scale (BPNSFS; [13], which was originally developed by Chen and colleagues (2015) and is now available in 14 languages adapted for various contexts [17]. Numerous validation studies indicated evidence for a six-factor solution of this scale [4,9,13], referring to the satisfaction or frustration of one of the three basic psychological needs with each factor. Some studies also found support for higher-order models, thereby creating composite need scores, with the risk of overlooking need-specific effects [13,18].

To further validate SDT’s universality hypothesis and to add to the emerging literature on the transdiagnostic role of basic psychological needs, this study aimed to 1. validate the BPNSFS as an assessment in the clinical context by examining the role of the basic needs in individuals with depression, and 2. develop a short version of the German BPNSFS. This is an important mission given the need for brief and valid assessments in both clinical and health service research contexts where time is of concern (e.g., lower psychological stress for clinical samples, lower dropout rates). It is conceivable that the questionnaire could be used to evaluate health treatments that target psychological health (e.g., psychotherapy or exercise therapy). Heissel et al. [19] use the questionnaire to explore rumination as a possible coping strategy to deal with experienced need frustration and a recent article suggests that the basic need questionnaire could also be practical in deriving specific treatment strategies if the respective profile is known [20]. Short versions of basic need instruments have proven to be valid and reliable instruments [21], thereby representing a good and economical alternative to the long versions. To do so, four different hypothetical models were tested: (1) a 3-factor model; (2) a 6-factor model [13]; (3) a 2-factor hierarchical model (6^2^-factor); and (4) a 3-factor hierarchical model (6^3^-factor) (Figure A2, Appendix B). It was hypothesized that the data would show a better fit for the 6-factor solution when compared to the 3-factor and the 6^3^-factor hierarchical model and would show an acceptable fit for the 6^2^-factor hierarchical model, indicating that the use of composite scores of need satisfaction or need frustration would be feasible. It was further hypothesized that the 6-factor model would show an acceptable to good fit for the subsamples with different severities of depressive symptoms and that measurement invariance could be established. Following this, the predictive validity of the 12-item version of the German BPNSFS was investigated and it was expected that need frustration especially relates to ill-being (depressive symptoms and anxiety) and need satisfaction mostly relates to physical and mental quality of life.

## 2. Materials and Methods

### 2.1. Participants and Procedure

Data from the baseline assessment of the project “STEP.De -Sports Therapy for Depression”, assessing the implementation of sports therapy as an alternative treatment in depressed patients [22], were used. Patients were recruited through four local health insurance carriers in Berlin, Germany, between August 2018 and March 2021. Strict inclusion and exclusion criteria were followed for the clinical sample of mild to moderately severe depression diagnosed by psychotherapists using the Structural Clinical Interview I for Diagnostic and Statistical Manual of Mental Disorders 4 (DSM IV), Structured Clinical Interview for DSM (SCID) I, Axis 1, Section A, E, and I [23] (for full details, see [22]). The validation sample consisted of *N* = 344 patients (71.8% female) with mild to moderately severe depression (Table A1). The mean age was 47.5 years (range = 20–65, *SD* = 11.1). Figure A1 shows a population pyramid of the sample according to age and gender.

Most patients indicated that they were married or cohabiting (59.1%) or single (26.3%). With respect to the highest completed education level, 56.0% of the patients completed secondary school, and 34.9% had higher education. Finally, with regard to personal monthly net income, 57.5% of the patients earned between EUR 1000 and 2000 (middle income) and 32.1% earned more than EUR 2000 (high income) (Additional Information on measures in Section A.1). Regarding their employment status, 82.2% of the patients worked within the last three months. Finally, 97.0% of the participants spoke German as a first language.

The missing rate for the BPNSFS items was low (0 to 1.2%). The ethics committee of the Freie Universität Berlin (No.206/18) and the University of Potsdam (No.17/2018) approved the study registered under the trial registration number ISRCTN28972230. Informed written consent was obtained from all participants.

Power analyses were a priori calculated using the R package ‘SampleSize4ClinicalTrials’ [24] We report how we determined our sample size, all data exclusions (if any), all data inclusion/exclusion criteria, whether inclusion/exclusion criteria were established prior to data analysis (see [22]), all measures in the study, and all analyses including all tested models.

### 2.2. Measures

A short 12-item version of the previously validated German 24-item version of the BPNSFS [9] was used to assess basic psychological need satisfaction and frustration. Items were selected based on the best factor loadings found in the German validation of the scale [9], the intersection with the original English version and the Dutch version of the scale [13], and the clearest German wording by consensus within the researcher team. Four items, of which two measured satisfaction and two measured frustration (e.g., competence satisfaction “I am good at what I do”, competence frustration “I feel disappointed with many of my performances”), all of them rated on a 5 point Likert scale with the range from 1 (completely disagree) to 5 (completely agree) assessed the three basic needs [4].

As indicators of subjective ill-being, depressive symptoms were measured with the German version [25] of the 21-item Beck Depression Inventory (BDI-II) [26] (e.g., “I feel sad most of the time”). Depression severity was classified according to Beck et al. [26]. The reliability in the present study had a Cronbach’s α = 0.90″. Anxiety was measured with 5 items from the VDS90 (Verhaltensdiagnostik-System/behavioral diagnostic system) [27] (e.g., “I avoid anxiety-inducing situations as often as possible”). The reliability in the present sample had a Cronbach’s α = 0.73.

Health-related quality of life as an indicator of well-being was measured by the 12-item Short Form Survey (SF-12) [28] (e.g., “In general, would you say your health is: excellent, very good, good, fair, poor?”). For extended information please see Additional Information on measures in Section A.3.

### 2.3. Analytical Strategy

Analyses were performed using IBM SPSS (Version 27) and R (Version 1.2.5042). Four different hypothetical models were tested via confirmatory factor analysis (CFA): (1) a 3-factor model representing the needs for autonomy, competence, and relatedness (not differentiating between satisfaction and frustration); (2) a 6-factor model differentiating between autonomy satisfaction, competence satisfaction, relatedness satisfaction, autonomy frustration, competence frustration, and relatedness frustration [13]; (3) a 2-factor hierarchical model (6^2^-factor) with six first-order (same as the previous model) and two second-order factors (need satisfaction and need frustration) included; (4) and a 3-factor hierarchical model (6^3^-factor) including six first-order factors (same as the two previous models) and three second-order factors (autonomy, competence, and relatedness) (Figure A2, Section A.2). CFAs were conducted using the “lavaan” package in R [29].

To evaluate the 3-factor model and the 6^3^-factor hierarchical model, frustration items were recoded to enable the creation of composite latent scores per need (Figure A2). Model comparison (CFA), measurement invariance (multigroup CFA), and predictive validity (structural equation modeling, SEM) were tested. For extended information about the analytical strategy please see Additional Information on analytical strategy in Section A.4.

## 3. Results

### 3.1. Descriptive Statistics and Correlations

As displayed in Table 1, correlational analyses showed that the three variables of need satisfaction were positively correlated with one another, while being negatively related to the three need frustration variables, which were equally positively correlated. In addition, need frustration correlated positively with ill-being and negatively with quality of life, whereas need satisfaction showed an opposite pattern of correlations. The scores for all items ranged from 1 to 5. Skewness values ranged between −0.85 and 1.17, thereby indicating minimal to moderate skewness. Kurtosis values ranged between −1.10 and 0.24, with some items showing a slightly flat distribution. Interitem correlations ranged from 0.08 to 0.55 in absolute value.

Next, the effects of sociodemographic variables were explored using a MANCOVA with gender, education level, and net income as between-subjects variables, age as a covariate, and the outcomes (depressive symptoms, anxiety symptoms, physical quality of life, and mental quality of life) as dependent variables. The results of the MANCOVA indicated that neither age (*F* (4269) = 2.37, ns), gender (*F* (4269) = 0.73, ns), education level (*F* (8538) = 1.57, ns), marital status (*F* (12,711.1) = 0.85, ns), or net income (*F* (8538) = 1.09, ns) yielded a significant multivariate main effect. Therefore, the sociodemographic variables were not included in the SEM analyses.

### 3.2. Confirmatory Factor Analyses

Chi-square tests were significant for all of the models, indicating that the models did not show a perfect model fit. As displayed in Table 2, the 3-factor model showed a poor fit, whereas the other models (especially the 6-factor model) showed an adequate fit. The 6-factor model showed a significant better fit than all other models, and this difference in fit was also found to be meaningful (ΔCFI > 0.01 for all model comparisons).

Based on the superior fit of the 6-factor model, this model was adopted to continue the scale validation process. Table A2 shows the parameter estimates of the CFA presenting this model. Standardized factor loadings were above 0.50 as well as being significant at a *p* < 0.001 level, showing small robust standard errors (ranging from 0.04 to 0.08) [4]. By including residual item correlations between items 3 and 10, the assumption of conditional independence for further analyses was loosened. A similar wording of both items, which was indicated by modification indices (M.I. = 18.253), made the inclusion plausible.

### 3.3. Reliability, Convergent, and Discriminant Validity

The need satisfaction and frustration subscales showed an adequate internal consistency, with Cronbach’s alpha at 0.75 and 0.69, respectively. The reliability of the 4-item autonomy (0.52) and relatedness (0.63) subscales was low, but adequate for competence (0.75). The coefficient omega (ω) [30] was calculated to evaluate the reliability of the three satisfaction subscales and the three frustration subscales in the 6-factor model, as it was considered a better indicator than Cronbach’s alpha [31]. Cronbach’s alpha was also calculated for comparison with other studies. Only the values of competence satisfaction and competence frustration exceeded the 0.70 threshold, showing adequate coefficient omega and Cronbach’s alpha values (Table A3).

Convergent was determined by calculating Average Variance Extracted (AVE), which estimates the common variance between the indicators and their latent factors. Convergent validity was established when the values of AVE exceeded the cut-off of 0.50 [32], indicating that more than 50% of the variance of the construct is due to its indicators. The results showed that the values of AVE exceeded the cut-off value of 0.50 for competence satisfaction (0.55) and competence frustration (0.54), but not for the other four factors (Table A3).

The evaluation of discriminant validity was performed by using the Fornell and Larcker (1981) [33] method based on the comparison between the AVE of each factor with the shared variance between factors. Discriminant validity is supported if the AVE of each factor is higher than the squared correlations between the factor and all other factors in the model. Calculated squared correlations are shown in Table A4. According to the results, discriminant validity was demonstrated for competence frustration.

### 3.4. Measurement Invariance across Patients with Different Severities of Depression

Across patients with different severities of depression, the model and its measurement invariance were tested. Therefore, the sample was divided into two groups: minimal/mild and moderate/severe depressive symptoms [9,26]. The primary aim of the current study was to examine the unique contribution of psychological need frustration and need satisfaction in the prediction of adults’ mental well-being and ill-being in a heterogeneous sample of adults (*N* = 334; *M*_age_ = 43.33, *SD* = 32.26; 53% females). Prior to this, validity evidence was provided for the German version of the Basic Psychological Need Satisfaction and Frustration Scale (BPNSFS) based on Self-Determination Theory (SDT). The results of the validation analyses found the German BPNSFS to be a valid and reliable measurement. Further, structural equation modeling (SEM) showed that both need satisfaction and frustration yielded unique and opposing associations with well-being. Specifically, the dimension of psychological need frustration predicted adults’ ill-being. Future research should examine whether frustration of psychological needs is involved in the onset and maintenance of psychopathology (e.g., major depressive disorder). When calculating the 6-factor model for the two subsamples separately, fit indices remained good for the sample with minimal/mild depressive symptoms and acceptable to good for the sample with moderate/severe depressive symptoms (Table A5). Overall weak measurement invariance (imposing equality of factor loadings) across the two subsamples, could be established with the constrained model not differing significantly from the unconstrained model (ΔSBS-χ^2^ (6) = 12.14, *p* < 0.059), and the SRMR and RMSEA indices for the constrained and unconstrained models not being substantially different (∆CFI < −0.01, ∆SRMR < 0.03, ∆RMSEA < 0.015), with the exception of the CFI difference (although this value was only slightly higher than the recommended cutoff of −0.01 (∆CFI = −0.013) [4]. Strong measurement invariance, which additionally requires equality constraints on the corresponding item intercepts, could not be established because the constrained model differed significantly from the unconstrained model (ΔSBS-χ^2^ (6) = 13.81, *p* < 0.032), and the difference in CFI indices was found to be slightly larger than −0.01 (∆CFI = −0.015).

### 3.5. Predictive Validity

The use of composite scores of need satisfaction and need frustration as predictors was substantiated by an acceptable fit to the data of the 6^2^-factor hierarchical model [4]. Need frustration related positively to depressive symptoms and anxiety, while relating negatively to physical quality of life and mental quality of life. On the contrary differences in ill-being or quality of life were not predicted by need satisfaction. Residual correlations between e9 and e10 were included in the model after the inspection of the modification indices, and are justified as PCS-12 and MCS-12 are two subscales of the same questionnaire. The chi-square test for the model was significant (χ^2^ (26) = 72.88, *p* < 0.001), indicating that the model did not show a perfect model fit. The model showed good SRMR (0.039), CFI values (0.954), and an acceptable RMSEA value (0.072). Figure A2 displays an overview of the SEM model. To assess whether the high standardized estimate between need frustration and depressive symptoms could be due to shared variance of need satisfaction and need frustration in the outcome, the existence of multicollinearity was investigated using the Variance Inflation Factor (VIF). The VIF was 1.30 for need satisfaction and need frustration and indicated that multicollinearity was not present to a critical degree.

## 4. Discussion

An increasing amount of research has now indicated that while need satisfaction is essential for individuals’ well-being and striving, need frustration is imperative in explaining ill-being and even psychopathology including depressive symptoms [3]. There is a need for a shorter version of the BPNSFS to increase its applicability in the clinical context due to scanty research on the effects of need-based experiences in clinical samples [4].

In a validation of the 12-item BPNSFS, it was shown that both the scale is considered reliable and the reliability of the subscales’ need satisfaction and need frustration is acceptable [4]. Focusing on the six factors (i.e., type of need X satisfaction/frustration), reliabilities were quite low (with the exception of competence satisfaction and frustration). This is understandable as each of these factors only contained two items. Further, CFA analyses confirmed our hypothesis that the 6-factor model and the 6^2^-model (to a lesser extent) are the best-fitting models. Both models distinguish between the satisfaction and frustration dimensions of the three needs. Compared to the 3-factor model and the 6^3^-model, the two models showed a better fit, as the former two did not distinguish between satisfaction and frustration of the needs explicitly. In the 24-item version of the German BPNSFS [7] and in previous validation studies in other languages [11], as well as theoretical assumptions [27], the same results could already be obtained. For example, the results of previously recently published Norwegian [4], Arabic [34], and Italian [35] BPNSFS validation studies, also indicated evidence for a 6-factor model. Compared to the 6-factor model, the fit of the 6^2^-factor model was worse. Possibly, this expected result, based on the methodological and theoretical assumptions, may be due to the need-specific variance shared by the items of the different latent factors [7]. A worse fit for the hierarchical 6^2^-factor model compared to the 6-factor model was also found by Heissel et al. (2018) [7].

In previous international studies, a worse fit for the hierarchical models compared to the reduced 6-factor model, was also shown (e.g., [11,29]). Due to its acceptable fit and the distinction between need satisfaction and need frustration as well as the distinction between different dimensions, the 6^2^-factor model is nevertheless feasible [16].

Aiming to investigate whether the 6-factor model would be found in samples differing in the severity of depressive symptoms, we ran additional multigroup CFAs. Intercepts between groups were not equal; although, weak invariance was found overall. As would be expected, the group with moderate/severe depressive symptoms was found to experience less need satisfaction and more need frustration compared to the group with milder depressive symptoms.

Focusing on ill-being and quality of life as outcomes, the final aim of this study was to examine the predictive value of need satisfaction and need frustration as assessed by the brief version of the BPNSFS [4]. Need frustration (but not need satisfaction) was found to relate to depressive symptoms and anxiety (positively) and to physical and mental quality of life (negatively). These findings, therefore, indicate that need satisfaction and need frustration indeed represent distinct constructs and should be assessed separately. In line with previous work, need frustration, but not need satisfaction, was found to be essential in the prediction of ill-being [3]. Additionally, these results extend previous research by showing the importance of need-based experiences, and especially frustration experiences, in a clinical sample [12]. Surprisingly, and in contrast to the results obtained in previous studies [13,17], need satisfaction did not relate to quality of life once need frustration was accounted for. A possible explanation for this unexpected finding is related to the type of well-being indicators that were assessed in this study. That is, in contrast with previous research mainly focusing on the predictive value of need satisfaction in well-being indicators such as vitality, life satisfaction, or self-esteem (e.g., [13]), we focused on mental and physical quality of life. This quality of life was assessed with the Short Form Survey [28], which is quite strongly focused on limitations in daily life with only two items focusing on positive indicators of quality of life (i.e., feeling calm and peaceful, having a lot of energy). Given the predominant focus on maladaptive functioning within this questionnaire, it is understandable that especially need frustration was found to be predictive of quality of life. Another possible explanation for the absence of the predictive value of need satisfaction is related to the employed sample. In a clinical sample, it is possible that the extent of experienced need frustration is of stronger predictive value for indicators of both well-being and ill-being when compared to a more general sample. Further longitudinal research is needed to find out whether the absence of need frustration contributes additional valuable insights into human well-being beyond the presence of need satisfaction.

### Limitations

A limitation of this study is the small sample size (in each subgroup), which may have caused a poorer estimation of the multigroup analysis (see [36]). Furthermore, due to the cross-sectional design, causality assumptions cannot be made, which in turn limits the results of the structural analysis.

The six factors of the proposed model are composed of two items each, not fulfilling the general requirement of three items to saturate a factor [32]. However, the use of the few best indicators has also been recommended, considering that one or two indicators are often sufficient, and encourages development of theoretically sophisticated models. Therefore, the two factors are retained in the 6-factor model [37].

## 5. Conclusions

The main findings showed an expected pattern of results within a sample of clinically depressed people, with high basic need frustration associated with greater self-reported symptomatology. Perhaps more important, the results of this study also yielded an economical and valid short form of the BPNSFS questionnaire that is useful for health science research. For the 12-item short version of the German BPNSFS, a six-factor solution differentiating between the three needs (autonomy, competence, and relatedness) in two dimensions (frustration, satisfaction) was feasible. The BPNSFS questionnaire in its short form turned out to be a beneficial instrument for specific health groups. In terms of predictive validity in a depressed sample, the recently added dimension of need frustration was found to be distinct from the satisfaction dimension and the only predictive value for all of the health-related outcome variables. Future research should investigate these associations incorporating different ways of using the scales (e.g., calculate balanced scores according to dimension). If these results can be verified in longitudinal studies, the assessment of basic need frustration alongside basic need satisfaction becomes crucial not only in the context of psychological ill-being. The great potential for future research can result in a deeper understanding of the onset and maintenance of ill-being and add substantial evidence in support of the underlying theory [4].

## Figures and Tables

**Table 1 healthcare-11-00412-t001:** Means, standard deviations, and correlations among the main study variables.

Measure	*M*	*SD*	1	2	3	4	5	6	7	8	9	10	11	12
1. Need satisfaction ^a^	19.69	4.72	1											
2. Autonomy satisfaction	5.77	1.94	0.81 ***	1										
3. Competence satisfaction	6.19	2.09	0.80 ***	0.53 ***	1									
4. Relatedness satisfaction	7.71	1.92	0.73 ***	0.39 ***	0.37 ***	1								
5. Need frustration ^a^	15.79	4.78	−0.51 ***	−0.34 ***	−0.46 ***	−0.39 ***	1							
6. Autonomy frustration	6.51	2.04	−0.29 ***	−0.23 ***	−0.26 ***	−0.20 ***	0.68 ***	1						
7. Competence frustration	5.21	2.25	−0.46 ***	−0.34 ***	−0.49 ***	−0.25 ***	0.81 ***	0.38 ***	1					
8. Relatedness frustration	4.11	2.10	−0.41 ***	−0.20 ***	−0.31 ***	−0.44 ***	0.74 ***	0.24 ***	0.43 ***	1				
9. Depressive symptoms	22.63	9.89	−0.53 ***	−0.41 ***	−0.48 ***	−0.33 ***	0.64 ***	0.44 ***	0.60 ***	0.41 ***	1			
10. Anxiety symptoms	0.92	0.74	−0.26 ***	−0.17 **	−0.29 ***	−0.15 **	0.28 ***	0.14 *	0.23 ***	0.23 ***	0.41 ***	1		
11. Physical quality of life	43.67	9.14	0.14 ***	0.03	0.19 **	0.14 *	−0.22 ***	−0.23 ***	−0.20 ***	−0.10	−0.37 ***	−0.26 ***	1	
12. Mental quality of life	31.52	9.16	0.36 **	0.28 ***	0.35 ***	0.17 **	−0.54 ***	−0.39 ***	−0.46 ***	−0.33 ***	−0.65 ***	−0.30 ***	0.01	1

Sample size ranged from *n* = 305 to *n* = 344 due to missing values in the variables. ^a^ Composite score over all three needs. * *p* < 0.05; ** *p* < 0.01; *** *p* < 0.001.

**Table 2 healthcare-11-00412-t002:** Goodness-of-fit indices of the tested models (*N* = 344).

	χ^2^	*df*	χ^2^/*df*	CFI	TLI	SRMR	RMSEA (90% CI)	AIC	BIC	Model Comparison ΔSBS-χ^2^ (Δ*df*)
6-factor model	68.869 **	39	1.77 ^g^	0.963 ^g^	0.937	0.039 ^g^	0.049 ^g^ (0.029, 0.068)	12,536.838	12,570.926	
3-factor model	190.819 ***	51	3.74 ^n^	0.821 ^n^	0.768	0.068 ^a^	0.095 ^n^ (0.081, 0.109)	12,653.019	12,679.086	112.42 (12) ***
6^2^-factor hierarchical model	108.419 ***	47	2.31 ^a^	0.920 ^a^	0.888	0.051 ^a^	0.066 ^a^ (0.050, 0.082)	12,569.265	12,598.005	35.22 (8) ***
6^3^-factor hierarchical model	130.759 ***	45	2.91 ^a^	0.894 ^n^	0.845	0.060 ^a^	0.078 ^a^ (0.062, 0.093)	12,591.771	12,621.849	63.575 (6) ***

*df*, Degrees of Freedom; CFI, Comparative Fit Index; TLI, Tucker–Lewis Index; SRMR, Standardized Root Mean Square Residual; RMSEA, Root Mean Square Error of Approximation; AIC, Akaike’s Information Criterion; BIC, Bayesian Information Criterion; ΔSBSχ^2^, Satorra–Bentler scaled chi-square difference. ^g^ Good value; ^a^ acceptable value; ^n^ unacceptable value. Models were compared with the 6-factor model. ** *p* < 0.01; *** *p* < 0.001.

## Data Availability

The datasets generated for this study are available upon reasonable request to the corresponding author.

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
