# Peer review of "Validating the German Short Basic Psychological Need Satisfaction and Frustration Scale in Individuals with Depression"

_healthcare, 2023, doi:10.3390/healthcare11030412_

Round 1
Reviewer 1 Report
This is an excellent research. It is very complete and interesting. However, in a constructive spirit, I would like to make some small recommendations for improvement:
- It would be advisable to include a population pyramid when defining the study sample to visually show the distribution according to gender and age of the participants.
- It should be clarified why a 12-item version of the original test forated by 21 is used. It is supposed to be due to the adequacy of these items, having a significant MSA, but it is not clear why, for example, 12 and not a 10 or 14 version.
- Tables must appear immediately after when they are cited. For example, Table 1 appears in section 3.2 and is named in 3.1.
Nothing more, I reiterate my congratulations for the quality of the manuscript and professionalism of the project, it is always a pleasure to read scientific manuscripts of such quality.
Reviewer 2 Report
Heissel and the group here report the validation of the German version of 24-item Basic Psychological Need Satisfaction and Frustration Scale (BPNSFS) and a short 19 version (12 items) of this scale in individuals with depression. Based upon their confirmatory and validity analyses, authors propose that the reported short version of the BPNSFS is similarly reliable and fits a six-factor structure, and found that the need frustration is positively related to depressive symptoms and anxiety, whereas it was negatively related to the physical quality of life and mental quality of life. Importantly, it highlighted the importance of need frustration in the prediction of psychopathology.
Authors have provided strong justification for their hypotheses and study design through a well written introduction, with proper discussion of data-results with suitable figures/tables. Authors have done a great job of elaborating the key concept with proper and accurate examples in the parentheses, few examples if those are at lines 39, 40, 53, 54, 62, 63, 322. Authors can make a similar useful addition for readers between lines 99-100 highlighting the clinical applications where such short scale would be valuable in providing reliable and accurate predictions.
There are certainly few limitations which authors have identified and attempted to address in the discussion (e.g., small sample size, cross-sectional design, and an expected pattern of results within a sample of clinically depressed people). As highlighted by authors as well, validation of such a short scale would certainly require additional studies, preferably, longitudinal ones to also help establish causality conclusions, and multi-cultural or -country. An interesting study group for longitudinal study would be freshman college/university students belonging to academic or sports field.
Overall, the current manuscript is presented in a well-structured manner. Authors have cited appropriate and recent references through the manuscript. This manuscript has potential to contribute to the literature aiming to shorten many of the psychological scales to increase their usefulness in clinical setup.
I do not have any major comments or revisions to suggest.
Reviewer 3 Report
- The authors could state that the current study is the German validation study of the related scale in the title and the abstract.
- The introduction could include some information about the psychological pain (psychache) theory associated with unmet personal meets. The authors could benefit from this recent article: https://pubmed.ncbi.nlm.nih.gov/36425740/Why did the authors prefer to use the DSM-IV instead of the DSM-5?
- Were all the patients diagnosed with major depressive disorder according to DSM-IV criteria? Were any patients diagnosed with bipolar disorder depressive episode or schizoaffective disorder depressive type?
- The authors could present the results of their power analyses.
- What do the authors mean by mild or moderately severe depression? Please add a reference.
- Did any of the patients have psychotic features which could affect the answers to self-report scales?
- The authors could present the Cronbach alpha values of their sample for all Likert-type scales.
- The authors could present their sample's descriptive statistics and sociodemographic values in more detail.
- The statistical analyses seem appropriate; however, the authors should add a reference for mild/moderate depression classification to make the discriminant analyses meaningful.
- The authors could discuss their results with the validation studies of the related scale in different languages. The authors could also mention the languages. For example, I would like to see sentences like 'In the French validation study, …'
- Best regards
Round 2
Reviewer 3 Report
Authors responded my comments adequately . I have no further comments.